# Brazilian Validation of the European Organisation for Research and Treatment of Cancer (EORTC) Quality of Life Group (QLG) Computerised Adaptive Tests (CAT) Core

Gustavo Nader Marta [1,2,*], Tomás Y. T. de Souza [1], Alice R. N. de Souza e Silva [1], Ana Paula A. Pereira [1], David R. Ferreira Neto [1], Rie N. Asso [1], Fabiana A. M. Degrande [1], Guilherme Nader-Marta [3], Maurício F. da Silva [2,4], Rafael Gadia [1], Samir A. Hanna [1], Bernhard Holzner [5], Everardo D. Saad [6] and Morten Aagaard Petersen [7]

[1] Department of Radiation Oncology, Hospital Sírio-Libanês, Rua Dona Adma Jajet 115, Sao Paulo 01308-050, Brazil; tomas.ytsouza@hsl.org.br (T.Y.T.d.S.); alice.rnssilva@hsl.org.br (A.R.N.d.S.e.S.); ana.papereira@hsl.org.br (A.P.A.P.); david.rferreira@hsl.org.br (D.R.F.N.); rie.nasso@hsl.org.br (R.N.A.); fabiana.amiranda@hsl.org.br (F.A.M.D.); rafael.gadia@hsl.org.br (R.G.); samir.hanna@hsl.org.br (S.A.H.)

[2] Latin America Cooperative Oncology Group (LACOG), Porto Alegre 90619-900, Brazil; mauricio.f.silva@ufsm.br

[3] Academic Trials Promoting Team, Institut Jules Bordet, l'Université Libre de Bruxelles (U.L.B), 1210 Brussels, Belgium; guilherme.nadermarta@bordet.be

[4] Radiation Oncology Unit, Santa Maria University Hospital (UFSM), Santa Maria 97105-900, Brazil

[5] Department of Psychiatry, Psychotherapy and Psychosomatics, University Hospital of Psychiatry II, A-6020 Innsbruck, Austria; bernhard.holzner@tirol-kliniken.at

[6] Dendrix Research, Sao Paulo 04534-000, Brazil; everardo@dendrix.com.br

[7] Palliative Care Research Unit, Department of Geriatrics and Palliative Medicine GP, Bispebjerg & Frederiksberg Hospital, University of Copenhagen, 2400 Copenhagen NV, Denmark; Aagaard.Petersen@regionh.dk

[*] Correspondence: gustavo.marta@hsl.org.br

**Abstract:** Background: This study aimed to validate the Brazilian version of EORTC CAT Core and compare the Brazilian results with those from the original European EORTC CAT Core validation study. Methods: After validated translation, 168 cancer patients from Brazil receiving radiation therapy with or without chemotherapy was assessed. Translated EORTC CAT Core and all QLQ-C30 items were administered to patients using CHES (Computer-Based Health Evaluation System) before (T0) and after (T1) treatment initiation. The association between QLQ-C30 and CAT scores and ceiling/floor effects were estimated. Based on estimates of relative validity (cross-sectional, known-group differences and changes over time), relative sample-size requirements for CAT compared to QLQ-C30 were estimated. Results: Correlation coefficients between CAT and QLQ-C30 domains ranged from 0.63 to 0.93; except for dyspnoea, all coefficients were >0.82 (corresponding figures were 0.81–0.93 in the European study). On average across domains, floor/ceiling was reduced by 10% using CAT (9% in the European study) corresponding to a relative reduction of 32% (37% in the European study). Analyses of known-group validity and responsiveness indicated that, on average across domains, the sample-size requirements may be reduced by 17% using CAT rather than QLQ-C30, without loss of power (28% in the European study). The Brazilian sample had less symptom/quality of life impairment than the European sample, which likely explains the lower sample-size reduction using CAT when comparing with the European sample. Conclusions: The results in the Brazilian cohort were generally similar to those from the European sample and confirm the validity and usefulness of the EORTC CAT Core.

**Keywords:** health-related quality of life; treatment; patient-reported outcome measures; computerized adaptive test

## 1. Introduction

The patient-reported outcome measures (PROMs) evaluating health-related quality of life (HRQoL) are usually not assessed using dynamic questionnaires. With the instruments typically used, an identical set of items structured in the same sequence is presented to all persons in order to ensure that results are comparable among patients. The European Organisation for Research and Treatment of Cancer (EORTC) Quality of Life Group has developed computerized adaptive tests (CATs) covering the 14 symptom and functional domains of the EORTC Quality of Life Core Questionnaire, the QLQ-C30 [1]. The EORTC CAT Core was developed to measure the same domains as the QLQ-C30 but with improved measurement accuracy, relevance to patients, and flexibility [2–7]. Another possible advantage is that, being an instrument with improved measurement precision, CAT requires a relatively smaller patient sample when compared with a static questionnaire while maintaining an equivalent level of statistical power [4,5,8–10].

The development and the comprehensive results of specific EORTC CAT item banks have been reported by the EORTC Quality of Life Group. Of note, initial validation studies during the construction of the item banks showed that CAT measures conceptually are similar and comparable to the QLQ-C30 instrument [4–13].

Recently, Petersen et al. reported the first international validation of the EORTC CAT Core, including 699 cancer patients from seven European countries. Based on this study, it was demonstrated that CAT, when compared with the QLQ-C30, largely simplified the assessment of these PROMs with the use of smaller samples and no loss of power, thus being a suitable instrument for general use [8].

Despite all this evidence in favour of the EORTC CAT Core utilisation, before its wider use globally, it is important to generate further evidence to support the use of this instrument. This study aimed to validate the Brazilian version of the EORTC CAT Core and compare the Brazilian results with those from the European EORTC CAT Core validation study.

## 2. Methods

### 2.1. Instruments and Translation Procedure

The EORTC CAT Core has been described previously [13]. In brief, it consists of 14 item banks covering the symptom and functional domains of the QLQ-C30 (see the list of domains in Table 1). Each item bank comprises between 7 and 34 items, with a total of 260 items. The CAT scores are on a so-called T-score metric, scaled so that the European general population has a mean of 50 and a standard deviation of 10 for all domains [14].

The EORTC CAT Core was translated to Portuguese according to the EORTC Quality of Life Group recommendations. All the translation phases (preparation stage, forward translations, reconciliation, back translations, back translation report, proofreading, pilot-testing, and final translation) were performed [15].

The EORTC QLQ-C30 consists of 30 items covering the same 14 domains as the EORTC CAT Core plus overall health/quality of life. The QLQ-C30 scales range from 0 to 100 and were scored according to the official QLQ-C30 scoring manual [16]. For both CAT and QLQ-C30, a higher score indicates better functioning for the functional domains and worse symptoms for the symptom domains.

### 2.2. Study Design and Selection Criteria

To allow direct comparison of results of the current study with those of the original European validation of the EORTC CAT Core, the study design and inclusion criteria followed those of the original validation study [8].

Patients from one Oncology Centre in Sao Paulo (Brazil) were included in this prospective survey. The inclusion criteria used in the study were: age 18 years or older; histological confirmation of cancer; planned radiation therapy and/or systemic therapy; having Portuguese as native language; and having the capacity to understand and complete the EORTC CAT Core.

**Table 1.** Brazilian data. The EORTC CAT Core and QLQ-C30 scores at baseline (T0) and follow-up (T1) for Brazilian participants.

| Domain | N | Mean | Standard Deviation | Median | Minimum | Maximum |
|---|---|---|---|---|---|---|
| Baseline (T0) | | | | | | |
| EORTC CAT Core | | | | | | |
| AP | 168 | 50.2 | 9.6 | 43.9 | 43.9 | 87.8 |
| CF | 168 | 51.5 | 10.4 | 53.2 | 15.4 | 62.3 |
| CO | 168 | 49.8 | 11.0 | 41.1 | 41.1 | 87.3 |
| DI | 168 | 47.2 | 8.1 | 43.8 | 43.8 | 92.2 |
| DY | 168 | 47.1 | 8.0 | 44.0 | 40.4 | 82.0 |
| EF | 168 | 50.5 | 8.9 | 51.1 | 19.2 | 63.9 |
| FA | 168 | 44.9 | 9.4 | 43.3 | 34.3 | 82.1 |
| FI | 168 | 53.4 | 9.7 | 50.6 | 42.1 | 87.4 |
| NV | 168 | 48.9 | 8.1 | 45.8 | 45.8 | 106.1 |
| PF | 168 | 45.2 | 9.2 | 45.5 | 9.8 | 66.3 |
| PA | 168 | 46.0 | 9.1 | 39.4 | 39.4 | 78.7 |
| RF | 168 | 48.2 | 10.3 | 48.8 | 16.5 | 58.8 |
| SF | 168 | 48.5 | 9.4 | 50.4 | 17.7 | 57.6 |
| SL | 168 | 50.1 | 10.1 | 48.6 | 36.9 | 75.6 |
| QL | 168 | 62.0 | 12.8 | 67.2 | 20.0 | 77.4 |
| QLQ-C30 | | | | | | |
| AP | 168 | 11.5 | 22.8 | 0.0 | 0.0 | 100.0 |
| CF | 168 | 85.3 | 21.3 | 100.0 | 0.0 | 100.0 |
| CO | 168 | 14.3 | 26.2 | 0.0 | 0.0 | 100.0 |
| DI | 168 | 6.0 | 16.8 | 0.0 | 0.0 | 100.0 |
| DY | 168 | 6.0 | 17.6 | 0.0 | 0.0 | 100.0 |
| EF | 168 | 74.7 | 22.8 | 75.0 | 0.0 | 100.0 |
| FA | 168 | 18.7 | 21.0 | 11.1 | 0.0 | 100.0 |
| FI | 168 | 17.9 | 29.2 | 0.0 | 0.0 | 100.0 |
| NV | 168 | 5.0 | 12.8 | 0.0 | 0.0 | 100.0 |
| PF | 168 | 82.4 | 17.2 | 86.7 | 0.0 | 100.0 |
| PA | 168 | 15.1 | 25.0 | 0.0 | 0.0 | 100.0 |
| RF | 168 | 85.2 | 25.2 | 100.0 | 0.0 | 100.0 |
| SF | 168 | 81.2 | 26.2 | 100.0 | 0.0 | 100.0 |
| SL | 168 | 27.2 | 33.8 | 0.0 | 0.0 | 100.0 |
| QL | 168 | 74.4 | 21.7 | 83.3 | 0.0 | 100.0 |
| Follow-up (T1) | | | | | | |
| EORTC CAT Core | | | | | | |
| AP | 107 | 52.7 | 11.3 | 43.9 | 43.9 | 87.8 |
| CF | 107 | 52.3 | 10.0 | 54.1 | 22.7 | 62.3 |
| CO | 107 | 50.5 | 10.4 | 47.5 | 41.1 | 78.4 |
| DI | 107 | 49.3 | 9.8 | 43.8 | 43.8 | 82.9 |
| DY | 107 | 48.3 | 8.6 | 47.6 | 40.4 | 68.5 |
| EF | 107 | 53.6 | 7.6 | 55.6 | 34.7 | 63.9 |
| FA | 107 | 47.9 | 9.0 | 46.8 | 34.3 | 74.3 |
| FI | 107 | 52.9 | 9.4 | 50.6 | 42.1 | 87.4 |
| NV | 107 | 51.0 | 9.5 | 45.8 | 45.8 | 99.8 |
| PF | 107 | 44.6 | 7.7 | 45.3 | 26.5 | 66.3 |
| PA | 107 | 44.9 | 7.9 | 39.4 | 39.4 | 67.1 |
| RF | 107 | 47.4 | 8.7 | 46.0 | 27.5 | 58.8 |
| SF | 107 | 48.5 | 8.6 | 50.4 | 29.8 | 57.6 |
| SL | 107 | 47.7 | 9.9 | 43.1 | 36.9 | 75.6 |
| AP | 107 | 62.1 | 10.8 | 62.1 | 29.0 | 77.4 |
| QLQ-C30 | | | | | | |
| AP | 107 | 18.1 | 29.1 | 0.0 | 0.0 | 100.0 |
| CF | 107 | 87.2 | 19.4 | 100.0 | 0.0 | 100.0 |
| CO | 107 | 17.4 | 27.6 | 0.0 | 0.0 | 100.0 |
| DI | 107 | 10.6 | 23.6 | 0.0 | 0.0 | 100.0 |
| DY | 107 | 9.0 | 18.1 | 0.0 | 0.0 | 66.7 |
| EF | 107 | 83.6 | 16.9 | 91.7 | 25.0 | 100.0 |
| FA | 107 | 24.7 | 22.4 | 22.2 | 0.0 | 100.0 |

**Table 1.** *Cont.*

| Domain | N | Mean | Standard Deviation | Median | Minimum | Maximum |
|--------|---|------|--------------------|--------|---------|---------|
| FI | 107 | 17.8 | 27.6 | 0.0 | 0.0 | 100.0 |
| NV | 107 | 8.6 | 14.9 | 0.0 | 0.0 | 100.0 |
| PF | 107 | 82.9 | 15.1 | 86.7 | 33.3 | 100.0 |
| PA | 107 | 12.9 | 23.5 | 0.0 | 0.0 | 100.0 |
| RF | 107 | 81.2 | 23.6 | 83.3 | 16.7 | 100.0 |
| SF | 107 | 82.6 | 21.0 | 83.3 | 16.7 | 100.0 |
| SL | 107 | 18.7 | 29.4 | 0.0 | 0.0 | 100.0 |
| QL | 107 | 74.6 | 18.1 | 75.0 | 16.7 | 100.0 |

Note: AP, lack of appetite; CO, constipation; DI, diarrhoea; DY, dyspnoea; FI, financial difficulties; SL, insomnia; NV, nausea/vomiting; PA, pain; FA, fatigue; CF, cognitive functioning; RF, role functioning; SF, social functioning; EF, emotional functioning; PF, physical functioning; and QL, overall quality of life.

The EORTC CAT Core instrument was presented to the patients twice: at baseline (0 to 14 days before treatment initiation) and on follow-up (6 to 14 days after chemotherapy initiation or 1 to 8 days after completion of radiation therapy). At each assessment, patients completed CATs consisting of 5 to 9 items for each domain.

A web-based platform for collecting data was used. The EORTC CAT Core and all QLQ-C30 items were applied to patients using CHES (Computer-Based Health Evaluation System—https://ches.pro/index.php/ches, accessed on at the same time. All patients completed their assessments at the hospital where they underwent their oncological treatment using a personal computer (tablet or desktop).

### 2.3. Statistical Analyses

The statistical analyses generally followed those of the original validation study [8]. The applied analyses are summarised as follows. Further details about the statistical analyses are available from Petersen et al [8].

The association between QLQ-C30 and EORTC CAT Core scores were quantified through Pearson correlation coefficients, and the ceiling and floor effects were estimated.

To estimate the relative precision of the CAT scores compared to the QLQ-C30 scores, the ratio of the CAT score to the QLQ-C30 score information functions (a measure of precision [17]) were calculated.

The relative sensitivity, i.e., ability to detect expected cross-sectional group differences, of the EORTC CAT Core compared to the QLQ-C30 sum scales was evaluated by estimating relative validities (RVs). The RVs are calculated as the ratio of test statistics (TestCAT/TestC30) for testing expected group differences formed based on variables available at baseline T0. RV > 1 indicates greater power to detect differences using the CAT. Based on RV estimates, relative sample-size requirements for future studies using the EORTC CAT Core were calculated and compared with the corresponding requirements using QLQ-C30. A comparison of the Brazilian results with those from the European EORTC CAT Core validation study [8] was also performed. Although no formal sample-size calculation was performed for the current study, the initial aim was to have 100 evaluable patients for analysis (similar to the aim per country in the European validation); therefore, a larger sample size was required in order to account for potential loss to follow-up after the baseline assessment. SAS Enterprise Guide 7.1 software was used for analysis.

### 2.4. Ethics Approval

The study was approved by the ethics committee (Hospital Sírio-Libanês) and was performed in accordance with the ethical standards as laid down in the 1964 Declaration of Helsinki. Informed consent was obtained from all individual participants included in the study.

## 3. Results

### 3.1. Patient Characteristics

Between February 2018 and June 2019, 168 patients were included. Data were collected for all patients at baseline and for 107 (63.7%) patients on follow-up. Median age was 60.2 years, and the most common cancer type was breast cancer. All patients received both radiation therapy and systemic therapy. Table 2 presents the patient characteristics (sociodemographic and clinical) from the Brazilian cohort as well as selected information about the participants from the European EORTC CAT Core validation study [8].

**Table 2.** Clinical and sociodemographic characteristics at baseline of the included patients (Brazil) and for comparison in the original European validation study.

|  | Brazil N (%) * | European Countries N (%) ** |
|---|---|---|
| Baseline assessment | 168 (100.0%) | 694 (99.3%) |
| Follow-up assessment | 107 (63.7%) | 446 (63.8%) |
| Age, mean (SD) | 60.2 (13.8) | 60.6 (12.0) |
| Gender |  |  |
| Female | 91 (54.2%) | 391 (55.9%) |
| Male | 63 (37.5%) | 296 (42.4%) |
| Cancer stage |  |  |
| I–II | 83 (49.4%) | 207 (29.6%) |
| III–IV | 61 (36.3%) | 360 (51.5%) |
| Cancer site |  |  |
| Breast | 65 (38.7%) | 213 (30.5%) |
| Lung | 9 (5.4%) | 83 (11.9%) |
| Prostate | 29 (17.3%) | 45 (6.4%) |
| Ovary | 2 (1.2%) | 38 (5.4%) |
| Other | 48 (28.6%) | 256 (36.7%) |
| Treatment |  |  |
| Systemic therapy | 168 (100.0%) | 639 (91.4%) |
| Radiation therapy | 168 (100.0%) | 60 (8.6%) |
| Highest level of education |  |  |
| Less than compulsory | 2 (1.2%) | 27 (3.9%) |
| Compulsory | 10 (6.0%) | 178 (25.5%) |
| Post compulsory below university | 19 (11.3%) | 268 (38.4%) |
| University level or above | 121 (72.0%) | 185 (26.5%) |
| Household income |  |  |
| <40,000 euro | 47 (28%) | 206 (29.5%) |
| 40,000–79,999 euro | 49 (29.2%) | 63 (9.0%) |
| 80,000–119,999 euro | 20 (11.9%) | 32 (4.6%) |
| ≥120,000 euro | 10 (5.9%) | 16 (2.3%) |
| Do not know/want to disclose | 27 (16.1%) | 330 (47.3%) |
| Employment status |  |  |
| Retired | 42 (25.0%) | 280 (40.1%) |
| Working | 79 (47.0%) | 288 (41.2%) |
| Other | 32 (19.1%) | 91 (13.0%) |

Note: * Some percentages sum to less than 100% because of missing data; **: Results of the European study originally presented in: Petersen MA, Aaronson NK, Conroy T, Costantini A, Giesinger JM, Hammerlid E, et al. International validation of the EORTC CAT Core: a new adaptive instrument for measuring core quality of life domains in cancer. Qual Life Res. 2020;29:1405–17.

### 3.2. Brazilian Data

Table 1 summarizes the EORTC CAT Core and QLQ-C30 scores at T0 and T1 for Brazilian participants. The CAT scores at both T0 and T1 were close to the European general population mean of 50; most were within five points (0.5 SD). The only exception was the overall quality of life (QL), for which the Brazilian patients scored 12 points higher (better) than the European general population.

T1-T0 score differences for both instruments are displayed in Table 3. Positive differences indicate more symptoms or better functioning at T1. Differences reflecting better

scores (i.e., <0 for symptoms and >0 for functional scores) at T1 are highlighted in bold type. Generally, the changes from T0 to T1 were minor. For the EORTC CAT, the changes for all domains were within five points.

**Table 3.** Differences between follow-up (T1) and baseline (T0) scores of the EORTC CAT Core and QLQ-C30 for Brazilian participants.

| Domain | N | Mean | Standard Deviation | Median | Minimum | Maximum |
|--------|---|------|--------------------|--------|---------|---------|
| EORTC CAT Core | | | | | | |
| AP | 107 | 4.0 | 11.3 | 0.0 | −22.7 | 39.8 |
| CF | 107 | **1.2** | 8.9 | 0.0 | −21.7 | 25.3 |
| CO | 107 | 0.7 | 11.4 | 0.0 | −29.2 | 32.8 |
| DI | 107 | 2.1 | 10.1 | 0.0 | −26.9 | 39.1 |
| DY | 107 | 1.9 | 7.2 | 0.0 | −17.5 | 23.4 |
| EF | 107 | **3.0** | 7.1 | 2.3 | −19.4 | 19.4 |
| FA | 107 | 3.6 | 8.2 | 2.2 | −13.9 | 26.4 |
| FI | 107 | **−1.0** | 6.0 | 0.0 | −20.2 | 17.8 |
| NV | 107 | 2.7 | 9.6 | 0.0 | −30.6 | 37.1 |
| PA | 107 | **−0.1** | 9.0 | 0.0 | −19.3 | 27.6 |
| PF | 107 | −1.1 | 6.1 | −0.1 | −27.3 | 22.9 |
| QL | 107 | −0.8 | 13.1 | 0.0 | −39.2 | 37.2 |
| RF | 107 | −2.0 | 9.2 | 0.0 | −28.1 | 21.4 |
| SF | 107 | −0.6 | 8.7 | 0.0 | −26.0 | 27.8 |
| SL | 107 | **−2.6** | 9.7 | 0.0 | −38.7 | 27.5 |
| QLQ-C30 | | | | | | |
| AP | 107 | 10.3 | 31.2 | 0.0 | −66.7 | 100.0 |
| CF | 107 | **1.9** | 19.3 | 0.0 | −66.7 | 83.3 |
| CO | 107 | 2.2 | 30.1 | 0.0 | −100.0 | 100.0 |
| DI | 107 | 4.4 | 24.3 | 0.0 | −66.7 | 100.0 |
| DY | 107 | 2.8 | 20.0 | 0.0 | −66.7 | 66.7 |
| EF | 107 | **8.0** | 19.3 | 8.3 | −66.7 | 66.7 |
| FA | 107 | 7.9 | 20.8 | 0.0 | −55.6 | 88.9 |
| FI | 107 | **−0.6** | 19.4 | 0.0 | −66.7 | 66.7 |
| NV | 107 | 4.5 | 16.3 | 0.0 | −50.0 | 100.0 |
| PA | 107 | 0.8 | 25.3 | 0.0 | −66.7 | 100.0 |
| PF | 107 | −1.2 | 11.7 | 0.0 | −40.0 | 26.7 |
| QL | 107 | −1.2 | 22.2 | 0.0 | −66.7 | 66.7 |
| RF | 107 | −7.5 | 25.5 | 0.0 | −66.7 | 66.7 |
| SF | 107 | −1.1 | 26.0 | 0.0 | −66.7 | 100.0 |
| SL | 107 | **−8.7** | 35.9 | 0.0 | −100.0 | 100.0 |

Note: AP, lack of appetite; CO, constipation; DI, diarrhoea; DY, dyspnoea; FI, financial difficulties; SL, insomnia; NV, nausea/vomiting; PA, pain; FA, fatigue; CF, cognitive functioning; RF, role functioning; SF, social functioning; EF, emotional functioning; PF, physical functioning; QL, overall quality of life. Differences reflecting better scores (i.e., <0 for symptoms and >0 for functional scores) at T1 are highlighted in bold and color type.

The relative information precision of EORTC CAT Core compared with QLQ-C30 scores at and baseline (T0) is shown in the Table 4. Across the domains, the EORTC CAT Core provides mean = 5.5/median = 3.7 times as much information as the QLQ-C30 scores. Results for follow-up (T1) are similar to T0, with mean = 5.6/median = 3.7 times as much information across domains (details not shown).

*3.3. Comparison between Brazilian and European Data*

The comparison of QLQ-C30 scores at T0 between Brazilian and European participants are presented in the Supplementary Material Table S1. Overall, fewer symptoms/impairments were seen in the Brazilian sample, particularly regarding lack of appetite, dyspnoea, fatigue, role functioning, social functioning, and overall quality of life.

Table 5 demonstrates the correlations between EORTC CAT Core and QLQ-C30 scores and differences in correlation coefficients between Brazil and Europe. The relatively low

correlation for dyspnoea in the Brazilian sample is likely because most participants (87%) answered "not at all" to the QLQ-C30 item, but when asked about more demanding tasks (like walking up one flight of stairs), some (approximately 20%) responded "a little" or "quite a bit" dyspnoea. Hence, the relatively low correlation is mainly a result of a low sensitivity of the original item for individuals with little dyspnoea. Otherwise, the correlations between EORTC CAT Core and QLQ-C30 scores in the Brazilian sample were high and comparable to those found in the European cohort; for the comparisons between the two cohorts at T0, the median difference between correlation coefficients across all domains is <0.01.

**Table 4.** Brazilian data. Relative precision of the EORTC CAT Core compared to QLQ-C30 scores (information(CAT)/information(C30)).

| Variable | N | Mean | Standard Deviation | Median | Quartile Range | Minimum | Maximum |
|---|---|---|---|---|---|---|---|
| AP | 168 | 4.30 | 1.10 | 4.30 | 0.00 | 3.00 | 16.40 |
| CO | 168 | 5.80 | 2.40 | 7.10 | 3.50 | 2.80 | 20.50 |
| DI | 168 | 2.60 | 2.20 | 2.20 | 0.00 | 2.20 | 28.30 |
| DY | 167 | 19.00 | 7.80 | 25.70 | 14.10 | 4.30 | 25.70 |
| FI | 168 | 5.10 | 2.00 | 4.60 | 3.90 | 3.30 | 17.20 |
| SL | 168 | 13.10 | 16.10 | 4.20 | 8.20 | 3.20 | 47.70 |
| NV | 168 | 3.60 | 2.80 | 2.30 | 0.00 | 2.30 | 12.10 |
| PA | 168 | 2.30 | 1.10 | 1.50 | 1.90 | 1.50 | 7.80 |
| FA | 168 | 3.30 | 0.40 | 3.10 | 0.50 | 2.80 | 5.10 |
| CF | 168 | 4.50 | 1.10 | 4.20 | 2.50 | 2.90 | 5.80 |
| RF | 168 | 3.70 | 0.40 | 3.90 | 0.80 | 1.80 | 4.10 |
| SF | 168 | 2.40 | 0.80 | 2.00 | 0.20 | 1.70 | 7.00 |
| EF | 168 | 2.50 | 0.60 | 2.40 | 0.80 | 1.70 | 4.30 |
| PF | 168 | 5.00 | 1.40 | 4.80 | 2.20 | 2.80 | 8.00 |
| Across domains | | 5.5 | 6.8 | 3.7 | 2.8 | 1.5 | 47.7 |

Note: AP, lack of appetite; CO, constipation; DI, diarrhoea; DY, dyspnoea; FI, financial difficulties; SL, insomnia; NV, nausea/vomiting; PA, pain; FA, fatigue; CF, cognitive functioning; RF, role functioning; SF, social functioning; EF, emotional functioning; PF, physical functioning.

**Table 5.** Correlations between EORTC CAT Core and QLQ-C30 scores in Brazil and differences in correlations between Brazil and Europe.

| Domain | Brazil (T0) | Brazil (T1) | Brazil-Europe (T0) | Brazil-Europe (T1) |
|---|---|---|---|---|
| AP | 0.863 | 0.920 | −0.035 | 0.022 |
| CF | 0.867 | 0.880 | 0.007 | 0.002 |
| CO | 0.853 | 0.804 | −0.018 | −0.081 |
| DI | 0.900 | 0.897 | 0.020 | −0.003 |
| DY | 0.634 | 0.701 | −0.200 | −0.115 |
| EF | 0.878 | 0.795 | 0.005 | −0.060 |
| FA | 0.879 | 0.862 | −0.004 | −0.034 |
| FI | 0.821 | 0.875 | 0.007 | 0.058 |
| NV | 0.902 | 0.880 | 0.003 | −0.014 |
| PF | 0.866 | 0.864 | 0.007 | −0.039 |
| PA | 0.934 | 0.910 | 0.015 | −0.025 |
| RF | 0.837 | 0.783 | −0.032 | −0.125 |
| SF | 0.850 | 0.844 | −0.020 | −0.033 |
| SL | 0.879 | 0.908 | −0.004 | 0.010 |
| QL | 1.000 | 1.000 | 0.001 | 0.001 |
| Median | 0.867 | 0.875 | 0.001 | −0.025 |

Note: AP, lack of appetite; CO, constipation; DI, diarrhoea; DY, dyspnoea; FI, financial difficulties; SL, insomnia; NV, nausea/vomiting; PA, pain; FA, fatigue; CF, cognitive functioning; RF, role functioning; SF, social functioning; EF, emotional functioning; PF, physical functioning; QL, overall quality of life.

Overall, the floor/ceiling are on average reduced by 10% using EORTC CAT Core (9% in European study), corresponding to a relative reduction of 32% (37% in European). For

least symptoms/best functioning (i.e., floor for symptom domains and ceiling for functional domains), the reduction in Brazil is 18% (59% to 41%). The floor/ceiling reductions are generally similar in the Brazilian and European samples (Table 6).

**Table 6.** Floor and ceiling at T0 for the QLQ-C30 and the EORTC CAT Core, the differences in floor and ceilings between the two instruments, and the difference of this to the difference found in Europe.

|  | Brazil QLQ-C30 | Brazil CAT | Brazil CAT-QLQ-C30 | Brazil-Europe Difference |
|---|---|---|---|---|
| AP, floor | 75.0% | 66.1% | 8.9% | 0.2% |
| AP, ceiling | 2.4% | 0.6% | 1.8% | −2.1% |
| CF, floor | 1.8% | 1.8% | 0.0% | −0.6% |
| CF, ceiling | 53.6% | 32.7% | 20.8% | 0.2% |
| CO, floor | 71.4% | 52.4% | 19.0% | 1.6% |
| CO, ceiling | 4.2% | 1.2% | 3.0% | 1.1% |
| DI, floor | 86.9% | 82.7% | 4.2% | −1.6% |
| DI, ceiling | 0.6% | 0.6% | 0.0% | 1.0% |
| DY, floor | 86.9% | 50.0% | 36.9% | 14.0% |
| DY, ceiling | 1.8% | 0.6% | 1.2% | −1.3% |
| EF, floor | 1.8% | 0.6% | 1.2% | 0.5% |
| EF, ceiling | 17.3% | 13.7% | 3.6% | −0.6% |
| FA, floor | 32.7% | 23.2% | 9.5% | 6.0% |
| FA, ceiling | 0.6% | 0.6% | 0.0% | −4.1% |
| FI, floor | 66.7% | 23.8% | 42.9% | −0.6% |
| FI, ceiling | 5.4% | 0.6% | 4.8% | 2.3% |
| NV, floor | 81.0% | 78.0% | 3.0% | 0.7% |
| NV, ceiling | 0.6% | 0.6% | 0.0% | 0.0% |
| PA, floor | 62.5% | 57.1% | 5.4% | −1.9% |
| PA, ceiling | 3.0% | 0.6% | 2.4% | 0.9% |
| PF, floor | 0.6% | 0.6% | 0.0% | −0.1% |
| PF, ceiling | 20.8% | 4.2% | 16.6% | −5.0% |
| RF, floor | 1.8% | 1.2% | 0.6% | −2.3% |
| RF, ceiling | 66.1% | 35.7% | 30.4% | 16.3% |
| SF, floor | 3.0% | 1.8% | 1.2% | −5.8% |
| SF, ceiling | 52.4% | 39.3% | 13.1% | 4.0% |
| SL, floor | 53.0% | 17.3% | 35.7% | 4.8% |
| SL, ceiling | 8.9% | 1.2% | 7.7% | 4.6% |
| Mean | 30.8% | 21.0% | 9.8% | 1.1% |

Note: AP, lack of appetite; CO, constipation; DI, diarrhoea; DY, dyspnoea; FI, financial difficulties; SL, insomnia; NV, nausea/vomiting; PA, pain; FA, fatigue; CF, cognitive functioning; RF, role functioning; SF, social functioning; EF, emotional functioning.

Mean RV for each domain is shown in Supplementary Material Table S2. An extreme RV value was observed for dyspnoea in the Brazilian sample, reflecting that the C30 dyspnoea scale has low sensitivity in this sample, with few dyspnoea problems resulting in non-significant differences for the QLQ-C30 scale, while the CAT score results in significant differences. Otherwise, most differences found were small (8 of 14 RVs are <1.1), with a median RV across domains of 1.1, indicating average savings in sample-size requirements using the EORTC CAT Core versus QLQ-C30 of about 17%. Most RVs (10 of 14) are lower in the Brazilian sample than observed in the original European validation, and in the original validation, the average sample savings across domains were estimated to be 28%.

## 4. Discussion

Historically, information on patients' HRQoL comes from PROMs that are usually obtained using standardised, static questionnaires for which all responders need to answer the same set of items to create comparable scores. Of note, conventional PROMs frequently involve a considerable number of items, oftentimes more than might be reasonable to request patients to answer, to accomplish accurate measurements for patients at diverse levels of HRQoL [18].

CATs have the exceptional feature that permits the questionnaire to be adjusted to the individual patient without impairing the measurement of scores and the direct comparison across patients. For each patient, CATs select the most relevant items based on answers to previous items and can thereby optimise the measurement assessment. In this way, CAT increases measurement accuracy by focusing on questions of relevance to individual patients; increases flexibility, since the instrument can be adapted to each study; and allows real-time feedback of results due to the features of automatic device reporting. [18,19]

Due to the clear benefits of CAT, the EORTC Quality of Life Group have been developing the CAT PROM for cancer patients since 2005. Considering that EORTC QLQ-C30 is one of the most used instruments worldwide both for clinical assessment and cancer research, much effort has been put into developing the psychometric properties of this novel instrument [18,20–22]. In the studies that led to the development of CAT, collection and calibration of data for all domains evaluated by the EORTC QLQ-C30 allowed respecting the conceptual framework of the EORTC QLQ-C30, guaranteeing the highest possible compatibility with and ability to replace the original tool. Therefore, results of studies using EORTC CAT Core and the EORTC QLQ-C30 are comparable to each other. The EORTC CAT Core consists of 14 item banks, each including 7 to 34 items, resulting in a total of 260 items. This instrument was created considering different aspects, including interviews with cancer patients, expert recommendations, literature searches, and psychometric analyses. Furthermore, the EORTC CAT Core was developed transnationally, incorporating different cultures and languages, which supports it as an appropriate instrument across patient groups [1,18,23].

Despite the enthusiasm for developing the EORTC CAT Core that may have more accurate measurement compared with standard instruments, such as the QLQ-C30, and the existing data supporting this prospect, the efficacy of EORTC CAT Core benefits from confirmation in independent cohorts. The international validation study assessed the psychometric properties of the EORTC CAT Core in an independent sample and confirmed that the EORTC CAT Core has higher precision and measures the same HRQoL domains as the QLQ-C30 in the European perspective [8].

The present study is the first to address the evaluation of EORTC CAT Core in Latin America (Brazil). After being translated into Portuguese according to the EORTC Quality of Life Group recommendations, the EORTC CAT Core was prospectively applied to cancer patients. The Brazilian sample generally had less symptoms/impairment compared with the European sample. Of note, the Brazilian study was conducted in a privately held cancer centre, and perhaps for this reason, participants were generally wealthier and had a higher education level than was reported for the European sample. Additionally, the Brazilian patients generally deteriorated less from baseline T0 to follow-up T1. Nevertheless, the results in the Brazilian sample are comparable to the findings in the European sample except for a few differences, generally confirming the validity and applicability of the EORTC CAT Core in a non-European setting as well. Importantly, most correlations with QLQ-C30 and floor/ceiling are similar in comparison with the results from the European study. Likewise, for the estimates of relative validity of EORTC CAT Core compared with those from QLQ-C30, they were generally slightly lower in the Brazilian sample than among patients from Europe, which is in line with the former sample having less symptom/impairment and the fact that EORTC CAT Core particularly improves measurement for patients having symptom/impairment.

Our study has certain limitations. All patients are from the same centre, and it would be valuable to expand the validation to other centres in Brazil. However, as a strength, since the study used a broad mixed sample of patients (in gender, different ages, different cancer stage, etc.), our findings probably well represent the Brazilian population.

In conclusion, overall, the validation indicates that the EORTC CAT Core works well in the Brazilian sample and is comparable to the performance in the European sample.

**Supplementary Materials:** The following are available online at https://www.mdpi.com/article/10.3390/curroncol28050291/s1, Table S1: QLQ-C30 mean scores at T0 in Brazil and the difference

between Brazilian and European mean scores. Table S2: Mean relative validity/sensitivity (ability to detect cross-sectional differences) of the EORTC CAT Core compared to the QLQ-C30.

**Author Contributions:** All authors contributed to the study conception and design. Material preparation, data collection, and analysis were performed by G.N.M., T.Y.T.d.S., A.R.N.d.S.e.S., A.P.A.P., D.R.F.N., R.N.A., F.A.M.D., M.A.P., G.N.-M., M.F.d.S., E.D.S., B.H., R.G. and S.A.H. The first draft of the manuscript was written by G.N.M. and M.A.P. All authors commented on previous versions of the manuscript. All authors have read and agreed to the published version of the manuscript.

**Funding:** This research received no external funding.

**Institutional Review Board Statement:** The study was approved by the ethics committee (Hospital Sírio-Libanês 2.520.830) and was performed in accordance with the ethical standards as laid down in the 1964 Declaration of Helsinki.

**Informed Consent Statement:** Informed consent was obtained from all individual participants included in the study.

**Data Availability Statement:** The data that support the findings of this study are available from the corresponding author (G.N.M.) upon reasonable request.

**Conflicts of Interest:** All authors certify that they have no affiliations with or involvement in any organization or entity with any financial interest or non-financial interest in the subject matter or materials discussed in this manuscript. The authors have no financial or proprietary interests in any material discussed in this article.

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
