# Peer review of "Brazilian Validation of the European Organisation for Research and Treatment of Cancer (EORTC) Quality of Life Group (QLG) Computerised Adaptive Tests (CAT) Core"

_curroncol, doi:10.3390/curroncol28050291_

Round 1

Reviewer 1 Report

In the presented manuscript, the validation of the Brazilian version of EORTC CAT is presented and compared with the European validation study.

In general, the presentation is comprehensive and informative. Therefore, I confirm the authors' assessment that additional evidence for the use of the new instrument is provided here. After considering the following points, the manuscript is certainly suitable for publication.

Line 98: Why does Table 2 appear before Table 1? Referencing in the text is in the correct order.

Lines 108-125: The methodology describes that patients answered the EORTC CAT instrument. However, there is no indication of which patients answered the QLQ-C30 instrument and in what chronological order regarding EORTC CAT.

Line 143: Please justify why no sample size calculation was performed and why a sample size of 100 patients is considered sufficient.

Line 147: What software was used for analysis.

Line 183/186: The use of synonyms unnecessarily limits the readability of the manuscript. Please use terms consistently in the manuscript (here "relative information" vs. "relative precision").

Line 187: Please keep the order of items consistent across tables.

Line 187: The N indicates that the ratio was determined for baseline only. Please mention and justify this in the methodology.

Line 213: There is no explanation of floor/ceiling in the methodology. Please add this.

Line 230: The specific numbers for "savings in sample size requirements" per domain would be a valuable addition to Supplement 2.

Line 234-236: This is an interpretation which belongs in the discussion.

Lines 239-274: These sections are mostly a paraphrase of sections from the Introduction. Please remove all redundancy.

Lines 275-293: The actual discussion can still be substantially enriched and structured. Please use the following structure: addressing and interpreting the results, discussing the results and contextualizing them in the literature, limitations (incl. risk of bias) and strengths, deriving conclusions.

Author Response

In the presented manuscript, the validation of the Brazilian version of EORTC CAT is presented and compared with the European validation study.

In general, the presentation is comprehensive and informative. Therefore, I confirm the authors' assessment that additional evidence for the use of the new instrument is provided here. After considering the following points, the manuscript is certainly suitable for publication.

Line 98: Why does Table 2 appear before Table 1? Referencing in the text is in the correct order.

Thank you. We corrected the tables order.

Lines 108-125: The methodology describes that patients answered the EORTC CAT instrument. However, there is no indication of which patients answered the QLQ-C30 instrument and in what chronological order regarding EORTC CAT.

A web-based platform for collecting data was used. The EORTC CAT Core and all QLQ-C30 items were applied to patients using CHES. The patients answered the 2 instruments at the same time. This information was added in the text.

Line 143: Please justify why no sample size calculation was performed and why a sample size of 100 patients is considered sufficient.

This study overall aims to replicate the original CAT-validation study. For this a sample size of 100 per country was used as this was deemed feasible to collect and allowed for acceptable power to detect group difference within countries for the estimation of relative validity (with N=100 comparing two equal sized groups will provide power of 70-95% to detect median effect sizes of 0.5-0.8). This was deemed acceptable for the current study too.

Line 147: What software was used for analysis.

SAS Enterprise Guide 7.1 software was used for analysis. This information was added in the text.

Line 183/186: The use of synonyms unnecessarily limits the readability of the manuscript. Please use terms consistently in the manuscript (here "relative information" vs. "relative precision").

Thank you. We updated the text to use terms consistently in the article as suggested.

Line 187: The N indicates that the ratio was determined for baseline only. Please mention and justify this in the methodology.

The reviewer is correct that only results for T0 are presented. T1 provides highly similar results with overall across domains mean=5.6/median=3.7, hence, we do not think it is needed to show all results for T1 too.

Currently lines 183-185 states:

‘The relative information precision of EORTC CAT Core compared with QLQ-C30 scores is shown in the Table 4. Across the domains, the EORTC CAT Core provides mean=5.5/median=3.7 times as much information as the QLQ-C30 scores.’

Rephrased to:

‘The relative information precision of EORTC CAT Core compared with QLQ-C30 scores at and baseline (T0) is shown in the Table 4. Across the domains, the EORTC CAT Core provides mean=5.5/median=3.7 times as much information as the QLQ-C30 scores. Results for follow-up (T1) are similar to T0 with mean=5.6/median=3.7 times as much information across domains (details not shown)’

Line 213: There is no explanation of floor/ceiling in the methodology. Please add this.

As we described in the Methodology section, the information about statistical analyses (including floor/ceiling) issue were described in the European original validation study.

Petersen MA, Aaronson NK, Conroy T, Costantini A, Giesinger JM, Hammerlid E, et al. International validation of the EORTC CAT Core: a new adaptive instrument for measuring core quality of life domains in cancer. Qual Life Res. 2020;29:1405-17.

Since the statistical analyses are available from Petersen et al., we decided add the original reference only and not include more explanation of floor/ceiling in the text.

Line 230: The specific numbers for "savings in sample size requirements" per domain would be a valuable addition to Supplement 2.

This would be >300 numbers and the individual RVs are not of interest in themselves as they are subject to imprecision (it would be similar to presenting scores for individual patients in addition to sample means etc.). However, the reviewer has a point that the range of RVs may be of interest. Hence, we add min and max (yellow in the table below) to the Supplementary material 2 table.

Analysis Variable : rv_sign

var_cat

N Obs

N

Mean

Std Dev

Minimum

Maximum

T0_AP_CAT_score5

34

27

1.18

0.25

0.56

1.72

T0_CF_CAT_score6

34

23

1.07

0.23

0.63

1.71

T0_CO_CAT_score5

34

20

1.14

0.21

0.77

1.60

T0_DI_CAT_score5

35

6

1.27

0.38

0.75

1.62

T0_DY_CAT_score5

34

21

4.58

9.02

0.84

36.24

T0_EF_CAT_score8

34

26

1.04

0.21

0.60

1.60

T0_FA_CAT_score7

34

27

1.04

0.14

0.72

1.30

T0_FI_CAT_score5

36

20

1.38

0.75

0.64

4.33

T0_NV_CAT_score6

35

25

1.02

0.21

0.38

1.48

T0_PA_CAT_score6

34

28

1.01

0.14

0.75

1.30

T0_PF_CAT_score9

34

27

0.95

0.22

0.60

1.38

T0_RF_CAT_score6

34

27

1.11

0.23

0.74

1.75

T0_SF_CAT_score6

34

25

1.09

0.18

0.79

1.53

T0_SL_CAT_score5

34

23

1.17

0.21

0.86

1.46

Overall median:

Analysis Variable : mean_rv

Median

Minimum

Maximum

1.10

0.95

4.58

Line 234-236: This is an interpretation which belongs in the discussion.

That you. This information was excluded of the Results section.

Lines 239-274: These sections are mostly a paraphrase of sections from the Introduction. Please remove all redundancy. Lines 275-293: The actual discussion can still be substantially enriched and structured. Please use the following structure: addressing and interpreting the results, discussing the results and contextualizing them in the literature, limitations (incl. risk of bias) and strengths, deriving conclusions.

We respectfully disagree white the reviewer.  We interpreted the results and compared the findings to the European study, which we think is the only relevant comparison from the literature here. We added a new paragraph to describe the limitations of the study:

“Our study has certain limitations. All patients are from the same center and that it would be valuable to redo the validation other places in Brazil. However, as a strength that we have a broad mixed sample of patients (both gender, different ages, different cancer stage etc.), our finding probably well represent the Brazilian population.”

Reviewer 2 Report

In the manuscript by Nader Marta et al., titled, "Brazilian validation of the European Organisation for Research and Treatment of Cancer (EORTC) Quality of Life Group (QLG) computerised adaptive tests (CAT) Core," the authors investigate the EORTC CAT core in Brazil after being translated into Portuguese. The authors found that the European study and the Brazilian study were mostly congruent with each other. The only major differences were explained logically in the discussion section and this study validates that the EORTC CAT survey works well in the Brazilian cancer patient population. 

To improve the manuscript, I have a few suggestions. 

1) The first table that is introduced is table 2. Shouldn't it be table 1? 

2) I am a little hung up on the scaling that was performed for the European general population as described on page 2, lines 92-93. I don't have a problem with normalizing the mean for a comparison, but as soon as the authors fix the standard deviation to 10 for all domains, then that no longer becomes the European data and comparisons can not be accurately made. Why was the standard deviation scaled to 10 for all domains? 

3) The heading for the Brazil column in Table 1 needs to be fixed. Furthermore, I think a statistical comparison should be made between Brazil and the European countries. I think some important factors like cancer stage are borderline statistically significant, which if they are could skew some the the results to be more favorable for the Brazilian study, especially in support of statements like, "Only exception was the overall quality of life (QL) for which the Brazilian patients scored 12 points higher (better) than the European general population. Also in the footnote "per cents" has a space when it should be one word. 

4) In table 3, the authors highlight meaningful data in the table by bolding it. However, the difference is so subtle that I think the authors should bold and highlight the text with a color change. That way it will make it very easy for the reader to identify those differences. 

Author Response

Reviewer 2

In the manuscript by Nader Marta et al., titled, "Brazilian validation of the European Organisation for Research and Treatment of Cancer (EORTC) Quality of Life Group (QLG) computerised adaptive tests (CAT) Core," the authors investigate the EORTC CAT core in Brazil after being translated into Portuguese. The authors found that the European study and the Brazilian study were mostly congruent with each other. The only major differences were explained logically in the discussion section and this study validates that the EORTC CAT survey works well in the Brazilian cancer patient population. 

To improve the manuscript, I have a few suggestions. 

1) The first table that is introduced is table 2. Shouldn't it be table 1? 

Thank you. We corrected the tables order.

2) I am a little hung up on the scaling that was performed for the European general population as described on page 2, lines 92-93. I don't have a problem with normalizing the mean for a comparison, but as soon as the authors fix the standard deviation to 10 for all domains, then that no longer becomes the European data and comparisons can not be accurately made. Why was the standard deviation scaled to 10 for all domains? 

It seems we have not been clear in the description of the T-score scaling. It does not mean that the Brazilian sample has mean=50 and SD=10 (this is also evident from the scores presented in table 2). The scaling means that the scores obtained in the Brazilian sample can easily be compared to scores from the European general population as the European population have mean=50 and SD=10. For example, table 2 shows physical functioning, PF, in the Brazilian sample is 45.2 with SD=9.4, i.e., the Brazilian patients have a mean PF about 5 points lower than the European general population corresponding to about ½SD. Hopefully this clarifies what has been done. Note that such T-scoring relating scores to a reference population is often done, e.g. also by PROMIS which relate scores in a similar manner to the US general population.

3) The heading for the Brazil column in Table 1 needs to be fixed.

Thank you. The heading for the Brazil column in Table 1 was corrected.

Furthermore, I think a statistical comparison should be made between Brazil and the European countries. I think some important factors like cancer stage are borderline statistically significant, which if they are could skew some the results to be more favorable for the Brazilian study, especially in support of statements like, "Only exception was the overall quality of life (QL) for which the Brazilian patients scored 12 points higher (better) than the European general population.

 The current study only includes data from Brazil. Only summary measures describing the European sample included in the original validation has been added here to allow simple visual comparison for the reader to the European validation sample. Hence, no formal testing comparing is relevant or possible. The statement ‘Only exception was the overall quality of life (QL)…’ refers to comparison to the European general population (not the European validation sample of cancer patients) as is also stated in the beginning of the sentence ‘’ The CAT scores at both T0 and T1 were close to the European general population mean of 50…’. Hence, this simple uses to the interpretation of the T-scores (see response above) and does not relate to the European patient sample presented in table 1.’

Also in the footnote "per cents" has a space when it should be one word. 

Thank you. We corrected the word.

4) In table 3, the authors highlight meaningful data in the table by bolding it. However, the difference is so subtle that I think the authors should bold and highlight the text with a color change. That way it will make it very easy for the reader to identify those differences. 

Thank you. We updated the Table 3 as suggested. 

Round 2

Reviewer 2 Report

I am satisfied with the changes to the manuscript.